# Bright and stable anti-counterfeiting devices with independent stochastic processes covering multiple length scales

Junfang Zhang [1,2,3,5], Adam Creamer[1,2,3,5], Kai Xie[1,2,3], Jiaqing Tang[1,2,3], Luke Salter[1], Jonathan P. Wojciechowski [1,2,3] & Molly M. Stevens [1,2,3,4] ✉

Physical unclonable functions (PUFs) are considered the most promising approach to address the global issue of counterfeiting. Current PUF devices are often based on a single stochastic process, which can be broken, especially since their practical encoding capacities can be significantly lower than the theoretical value. Here we present stochastic PUF devices with features across multiple length scales, which incorporate semiconducting polymer nanoparticles (SPNs) as fluorescent taggants. The SPNs exhibit high brightness, photostability and size tunability when compared to the current state-of-the-art taggants. As a result, they are easily detectable and highly resilient to UV radiation. By embedding SPNs in photoresists, we generate PUFs consisting of nanoscale (distribution of SPNs within microspots), microscale (fractal edges on microspots), and macroscale (random microspot array) designs. With the assistance of a deep-learning model, the resulting PUFs show both near-ideal performance and accessibility for general end users, offering a strategy for next-generation security devices.

Counterfeiting is a serious global issue; the distribution of fake goods, such as electronics and medicines, causes a trillion-dollar economic loss every year, threatens national security and poses a danger to human health[1]. Physical unclonable functions (PUFs) offer a potential solution to this issue. During device fabrication, PUFs are generated by stochastic variations which can then be encoded into unique bit strings[2]. This intrinsic randomness makes PUFs practically immune to counterfeiting, even by the original manufacturer[3]. As a result, PUFs are one of the most promising anti-counterfeiting tools[4,5]. In practice, the manufacturer of the product would scan the PUF tags to generate 'challenge-response pairs' in cloud storage. The authenticator, upon receiving said product, would then scan (challenge) the PUF tag and cross-reference them to the cloud database to authenticate the product.

Silica-based PUFs are well-established and highly sought after due to the lack of external hardware (e.g. a microscope) needed for authentication[6,7]. However, the majority of silica-based PUFs show vulnerabilities to modelling attacks and suffer from a low reliability[6,8]. Optical PUFs offer an attractive alternative to silica due to their high entropy, high output complexity, and accessible readout approach. This is especially true for those engineered by chemical approaches as the inherent vast parameter space offers robust devices which are efficient for numerous applications[9–11].

Transition metal quantum dots (Qdots)[12], perovskite nanocrystals[13,14], organic dyes[15], and carbon dots[16,17] are the fluorescent taggants most commonly used in the development of optical PUF devices. More recently, silicon Qdots have also been shown as an alternative promising optical PUF taggant[18]. Semiconducting polymer nanoparticles (SPNs) are another viable taggant, as they have been shown to outperform most other fluorescent materials for both brightness and photostability[19,20]. The single-particle brightness of

[1]Department of Materials, Department of Bioengineering, Institute of Biomedical Engineering Imperial College London, London, UK. [2]Department of Physiology, Anatomy and Genetics, University of Oxford, Oxford, UK. [3]Kavli Institute for Nanoscience Discovery, University of Oxford, Oxford, UK. [4]Department of Engineering Science, University of Oxford, Oxford, UK. [5]These authors contributed equally: Junfang Zhang, Adam Creamer. ✉e-mail: molly.stevens@dpag.ox.ac.uk

SPNs has been reported to be more than thirty times higher than that of inorganic quantum dots and antibody-dye conjugates[21]. In addition, SPNs have been shown to have a superior photostability when compared to typical fluorescent dyes[22,23]. Recent works have illustrated semiconducting polymers in anti-counterfeiting devices, wherein polymer inks were printed in unique patterns[24] and engineered to show reversible colour changes in response to chemical and physical stimuli[9,25,26]. Similar approaches have also been shown with polymer-photoswitch conjugates in both paper and gels[10,11]. However, this photoswitch approach is susceptible to degradation upon repeated stimulation, especially when the process involves harsh conditions[15]. Furthermore, counterfeiters could copy the device by synthesizing the same underlying photo-active materials once they are discovered[27]. SPN-based PUF devices have yet to be explored and have the potential to outperform current fluorescent optical PUFs.

Herein, we present a chemically engineered PUF platform consisting of SPN taggants embedded in a patterned photoresist, achieved through stochastic processes. The platform guarantees both an ultrahigh security level with general accessibility to end-users at the same time. Specifically, we show a tunable size range of SPNs through solvent engineering, realizing a balance between the encoding

capacity and detection ability (Fig. 1a). The SPNs were found to be highly photostable, which allowed for patterning by photolithography without photodegradation. The obtained patterned photoresists were shown to be stable when immersed in 'artificial sweat' (formulated salt solutions), suggesting their potential for long-term use. A mixture of three orthogonal SPNs could facilitate red-green-blue (RGB) emission by a single excitation wavelength, due to the broad absorbance and large Stokes shift of the SPNs. Through exploration of the parameter space, we show that multiple stochastic processes (distribution of SPNs, fractal edges, and random microspot arrays) can be incorporated in a single high-security PUF device (Fig. 1b). The resulting PUF tags exhibited a near-ideal level of uniqueness, reliability, and bit uniformity. A deep-learning, open-source model (LoFTR, Detector-Free Local Feature Matching with Transformers)[28] was adjusted to achieve precise and fast identification (Fig. 1c). We then illustrate how these PUF devices can be printed on different substrates, demonstrating their high potential for applications in real-world products.

## Results and Discussion

We designed and optimized a semiconducting polymer-based anti-counterfeiting device. The device consists of a stochastic blend of

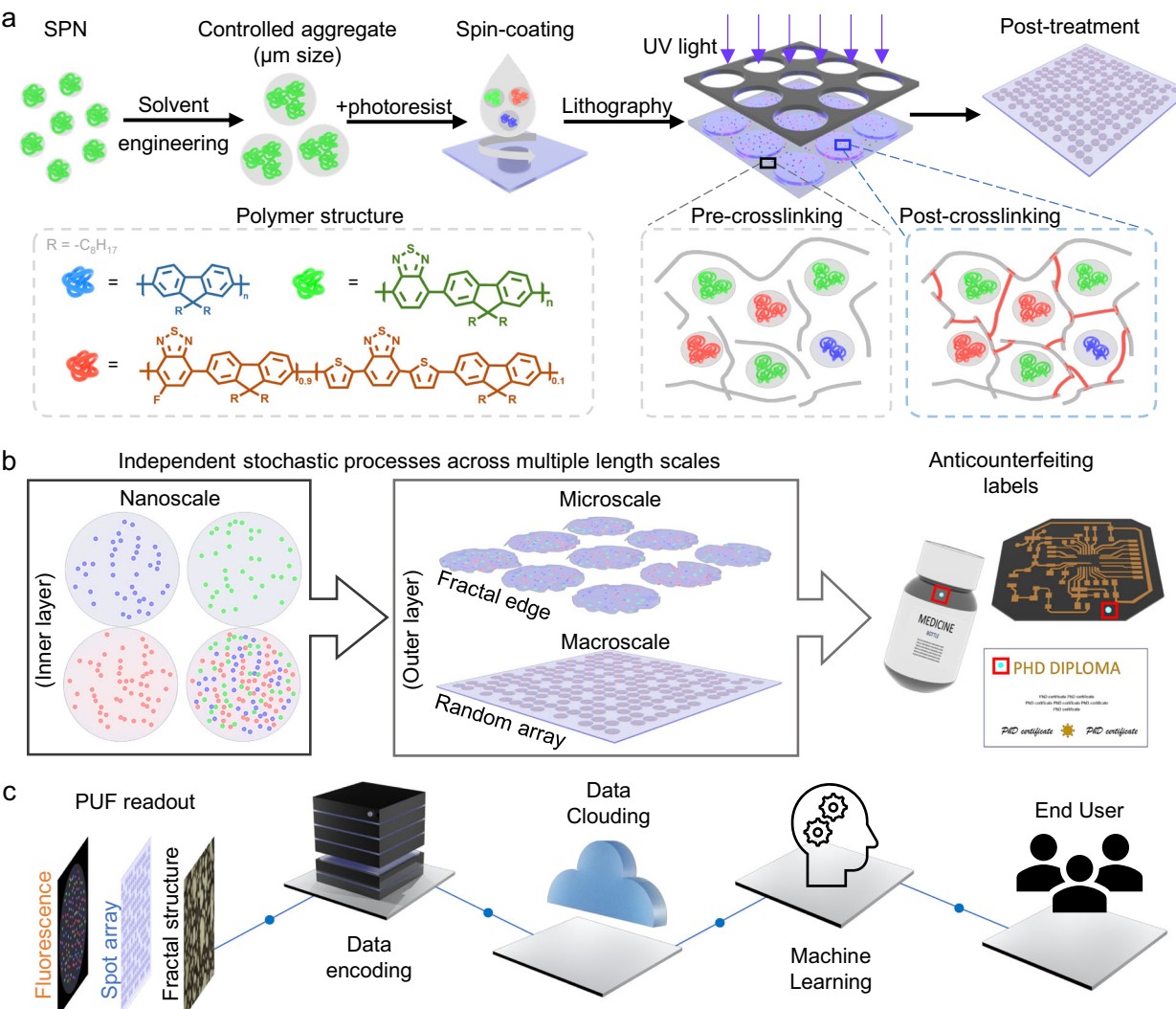

**Fig. 1 | Design of the PUF platform by material and process engineering. a** SPNs, with RGB emissions under the same wavelength excitation, are generated and aggregated in controlled sizes. The SPNs are mixed with photoresists and treated by a photolithography process. This cross-links the photoresists to improve the distribution stability of SPNs. **b** SPNs are randomly distributed in each microspot. Fractal edges and partially removed arrays are engineered during photo-lithography. The PUF devices can be applied as anticounterfeiting labels for different products. **c** Machine learning assisted identification process for end-user friendly security devices.

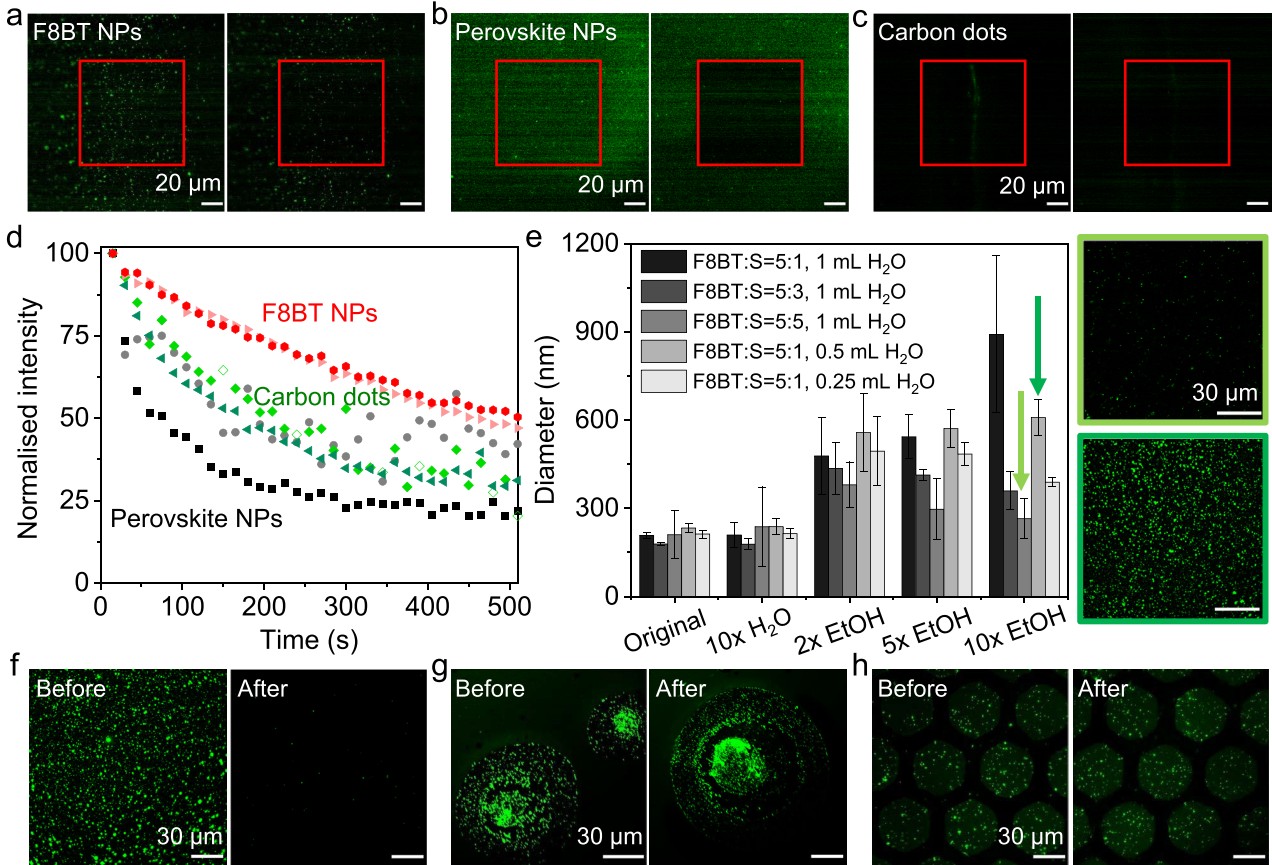

**Fig. 2 | Characterization and optimization of fluorescent NPs for PUF devices.** Confocal images of **a** F8BT (poly(9,9-dioctylfluorene-alt-benzothiadiazole) NPs (Ex: 405 nm, laser power: 15%, gain: 150%), **b** perovskite NPs (CsPbBr$_3$) (Ex: 405 nm, laser power: 15%, gain: 150%), and **c** carbon dots (Ex: 405 nm, laser power: 15%, gain: 500%), before and after photobleaching. The red square indicates in the area which was irradiated. **d** Photobleaching curves of nanoparticles imaged every 15 s under 80% laser power. Two shades of each colour indicate curves extracted from two separate regions of interest. **e** DLS analysis of F8BT NPs ($n = 3$, $N = 3$, S: surfactant, poly(styrene)-graft-poly(ethylene oxide)). Detailed results for each independent batch can be found in Fig. S8. The green arrows indicate nanoparticle batches which were cast as thin films, with their corresponding confocal images on the right (10% laser power and a gain value of 50% (Ex: 430 nm). Confocal images of **f** F8BT-PVA, **g** F8BT-PS, and **h** F8BT-SU-8 films before and after the incubation with artificial sweat (F8BT: Surfactant = 5:1, Ex: 430 nm, laser power: 10%, gain: 50%). Source data are provided as a Source Data file.

three SPNs with different distinct emission maxima which are embedded in a uniquely patterned transparent photoresist.

## Screening of fluorescent nanoparticles

We first explored the type of fluorescent nanoparticles (NP) to incorporate into the PUF device. The NPs were required to exhibit excellent brightness, photostability and a tunable emission wavelength. We opted to compare the brightness and photostability of materials through an experimental approach under identical conditions, as accurately comparing literature reported values with confidence is challenging. Therefore we initially screened SPNs (consisting of poly(9,9-dioctylfluorene-alt-benzothiadiazole (F8BT)) alongside carbon dots, Qdots (CdSe:ZnS), perovskite nanocrystals (CsPbBr$_3$), which have gained the most attention in the field as fluorescent taggants. We compared formulations of nanoparticles with emission maxima in the green (520–560 nm). F8BT was the semiconducting polymer of choice due to its prevalence in the literature, high reported brightness, photostability and good solubility in organic solvents[29].

All nanoparticles were immobilized onto glass slides, followed by confocal laser scanning microscopy (CLSM) and accelerated photobleaching. All the fluorescent materials were prepared at the same concentration of 0.01 g/L with the addition of a film-forming reagent, prior to spin-coating. We received the perovskite nanocrystals in toluene, therefore polystyrene (PS) was chosen as the film-forming additive. The remaining fluorescence materials were provided as aqueous dispersions, and as such polyvinyl alcohol (PVA) was used. After spin-coating, we obtained thin films embedded with fluorescent nanoparticles for confocal measurements. The excitation wavelength (405 nm) and laser power were consistent for all nanoparticles.

Under CLSM imaging, the fluorescent spots in the thin films were difficult to detect due to the small sizes of Qdots, carbon dots, and perovskite NPs. (Fig. 2a–c). In the case of Qdots, we found a few large particles (most likely caused by aggregation) but their intensity was two to ten times lower than the SPNs with comparable size (Fig. S1). We performed photobleaching tests under continuous laser scanning for 500 s. Images were taken before and after the photobleaching at two different areas for each sample (Fig. 2a–c, see Fig. S2 for additional images). We then extracted photobleaching curves from these images which quantitatively illustrated the superior photostability of F8BT NPs (Fig. 2d).

## Optimization and Characterization of SPNs for PUF devices

Given the high brightness and photostability of F8BT NPs, we opted to move forward in our study with SPNs. However, the initial particle sizes in the low nanometre ranges (10–100 nm diameter) were not ideal for practical readout due to their low signal-to-noise ratios. Initially, SPNs were synthesized following a common literature technique[30,31], in

which F8BT and a polystyrene-based surfactant (PS-PEG-COOH) in a 1:10 ratio (wt:wt) were blended in tetrahydrofuran (THF) before rapidly mixing with an excess of water (1:10, v:v). To achieve larger fluorescent spots, SPNs were formed with decreasing proportion of surfactant and decreasing relative water volumes. The resulting SPNs were then diluted into varying amounts of ethanol and the size distribution was measured by dynamic light scattering (DLS). Altering the relative surfactant and water content in the synthesis resulted in in a negligible change in size (Fig. 2e). However, when diluted into aqueous ethanol we observed a large size dependence. We found a higher amount of surfactant (F8BT:PS-PEG at 1:1) caused the particles to be more resistant to aggregation, even in a mixture with a 10-fold ethanol volume, maintaining a diameter of approximately 250 nm. Nevertheless, when we precipitated a lower surfactant ratio (5:1) into less water (1:5, THF:water), the nanoparticles aggregated to a consistent hydrodynamic diameter of approximately 600 nm, regardless of the ethanol content. The latter conditions produced much clearer, larger spots which could be imaged under a low-laser power (10%, Fig. 2e and Fig. S3). These conditions were chosen as the optimal parameters to produce for easy detectable and well reproducible SPNs. It is important to note that much larger aggregates were seen four hours after ethanol dilution (Fig. S4), thereby requiring thin-film formation shortly after the aggregate formation.

For the generation of PUFs, a facile and commonly used strategy is to spin-coat the fluorescent compounds within a polymer matrix. Initially, we chose PVA as the film-forming reagent to give SPN aggregate dispersions. The resulting thin-films were submersed in 'artificial sweat' for 30 min, to mimic the harsh environment that a practical device would need to endure. However, we observed a complete absence of SPNs (Fig. 2f). As a result, we then tested polystyrene (PS) as a water insoluble film-forming reagent (in which PS in dichloromethane is rapidly mixed with the SPN mixture, prior to spin-coating). Unfortunately, the SPNs exhibited further aggregation and significant changes in morphology when tested in sweat (Fig. 2g). To overcome this challenge, we introduced a UV cross-linking step to 'lock-in' the SPN aggregates by using a photoresist (SU-8) as the polymer matrix for spin-coating The high photostability of the SPNs (Fig. 2d) meant that photobleaching was not of concern during UV exposure. The resulting thin films showed an even distribution of aggregates which did not change after immersion in the sweat mimic (Fig. 2h).

## Multichannel PUF devices and encoding process

To increase the robustness of the PUF, we chose to develop a blend of SPNs with fluorescence in the blue, green and red channels (emission maxima at 440, 535 and 635 nm, respectively). For this purpose, optical devices often require multiple excitation wavelengths for their respective fluorescent constituents (due to their small Stokes shifts)[32] or require excitation with UV light, which typically accelerates bleaching. Semiconducting polymers are advantageous here as they typically exhibit a large Stokes shift[33]. Furthermore, due to the conjugated nature of semiconducting polymers, introduction of multiple chromophores into the polymer backbone can yield red and near-IR emitting polymers (by a cascade Förster energy transfer (FRET) mechanism)[34,35]. We chose two additional semiconducting polymers; poly(9,9-di-n-octylfluorenyl-2,7-diyl) (PFO) and poly[(9,9-dihexylfluorene)-co-2,1,3-benzothiadiazole-co-4,7-di(thiophen-2-yl)-2,1,3-benzothiadiazole] (F8BT-red) which have emission maxima in the blue and red, respectively (Fig. 3a). F8BT-red was synthesized (Fig. S5) based on previously reported literature[36]. Recapitulating the protocol developed for F8BT NPs, the optimisation process (surfactant ratio, water and ethanol content) was repeated for PFO and F8BT-red to yield larger particles (Fig. S6-7). The resulting nanoparticle formulations were then successfully embedded in an SU-8 matrix exhibiting three distinct colours under CLSM imaging (Fig. 3b, c).

The next step was to test the potential use of the three colour SPNs in anti-counterfeiting. Through excitation with incident light (henceforth described as a 'challenge') we generated a unique response that could subsequently function as a cryptographic key. The raw fluorescent images were converted to a binary bitmap[32] (Fig. 3d). To ensure high reproducibility, we use $10 \times 10$ pixels as a unit for binarization instead of just a single pixel (the size of a single pixel is $0.167\,\mu m \times 0.167\,\mu m$). In general, we found PFO SPNs to be smaller than both F8BT and F8BT-red NPs (Fig. S6). As such, we therefore united a larger area ($50 \times 50$ pixels) to enhance the reliability. We then combined three challenge-response pairs together with the final digitized key resulted in 216 bits per microspot (Fig. 3e). We collected a bitstream with a total size of 10,800 from 50 microspots, requiring only a small area, similar to that of a single mammalian cell. To assess the PUF performance, we subsequently characterized the bit uniformity, device uniqueness and readout reproducibility of the digitized keys.

Bit uniformity can be expressed by the Hamming weight of the bit states by the probability of observing either 1-bit or 0-bit. The ideal bit uniformity value is 0.5 in this case and is defined by the following equation:

$$bit\ uniformity = \frac{1}{k}\sum_{i=1}^{k} R_i \qquad (1)$$

where $R_i$ is the $i^{th}$ bit of the PUF pattern and $k$ is the total number of PUF patterns. Using this equation we calculated the bit uniformity from 100 different microspots (Fig. S8) to be $0.4948 \pm 0.0064$, indicating a high degree of randomness. The uniqueness of the PUFs is evaluated by inter-device Hamming distance (HD), which shows the number of different bits between two individual PUFs:

$$uniqueness = \frac{2}{k(k-1)}\sum_{i=1}^{k-1}\sum_{j=i+1}^{k} \frac{HD(R_i(n), R_j(n))}{n} \qquad (2)$$

where $R_i(n)$ and $R_j(n)$ are the $n$-bit responses of the $i^{th}$ and $j^{th}$ PUF patterns, respectively, and $k$ is the total number of the PUF patterns. The ideal uniqueness value for a PUF is also 0.5. Upon producing a histogram plot of the normalized inter-device HDs (Fig. 3f) and fitting to a Gaussian distribution, we observed a peak at $0.5000 \pm 0.0427$, revealing the high uniqueness of the microspot arrays. The reproducibility of the PUF responses can be assessed by calculating the intra-device HD when same PUF tag is challenged:

$$reproducibility = 1 - \frac{1}{k}\sum_{i=1}^{k}\frac{1}{T}\sum_{l=0}^{T} \frac{HD(R_i^0(n), R_i^l(n))}{n} \qquad (3)$$

where $R_i^l(n)$ is the $n$-bit response from the $i^{th}$ PUF at the $l^{th}$ trial, $k$ is the total number of PUF patterns and $T$ is the number of trials. In an ideal PUF system, the reproducibility is 100 for a stable acquisition. The estimated intra-device HD we obtained from 100 microspots has a mean value of $0.05568 \pm 0.0181$. Therefore, we can conclude the obtained security keys from the same PUF device exhibit high readout reproducibility and stability. We then utilized the intra-device HD (reproducibility) and inter-device HD (uniqueness) values to estimate the probabilities of an authentication error (AE) and false authentication (FA), as a function of the decision threshold (Fig. 3g). The FA probability provides an estimate of the probability of successful PUF cloning. At a decision threshold of 0.189, we found the probability of cloning to be below $10^{-12}$ for the SPN-loaded microspots. Furthermore, an open-source machine learning model, detector-free local feature matching with transformers (LoFTR)[37], was introduced for efficient PUF identification. We analysed the images obtained from CLSM by LoFTR to visualize the intra- and inter-correlations, offering a pairwise comparison map of cross-HD (Fig. 3h). The diagonal line illustrates the

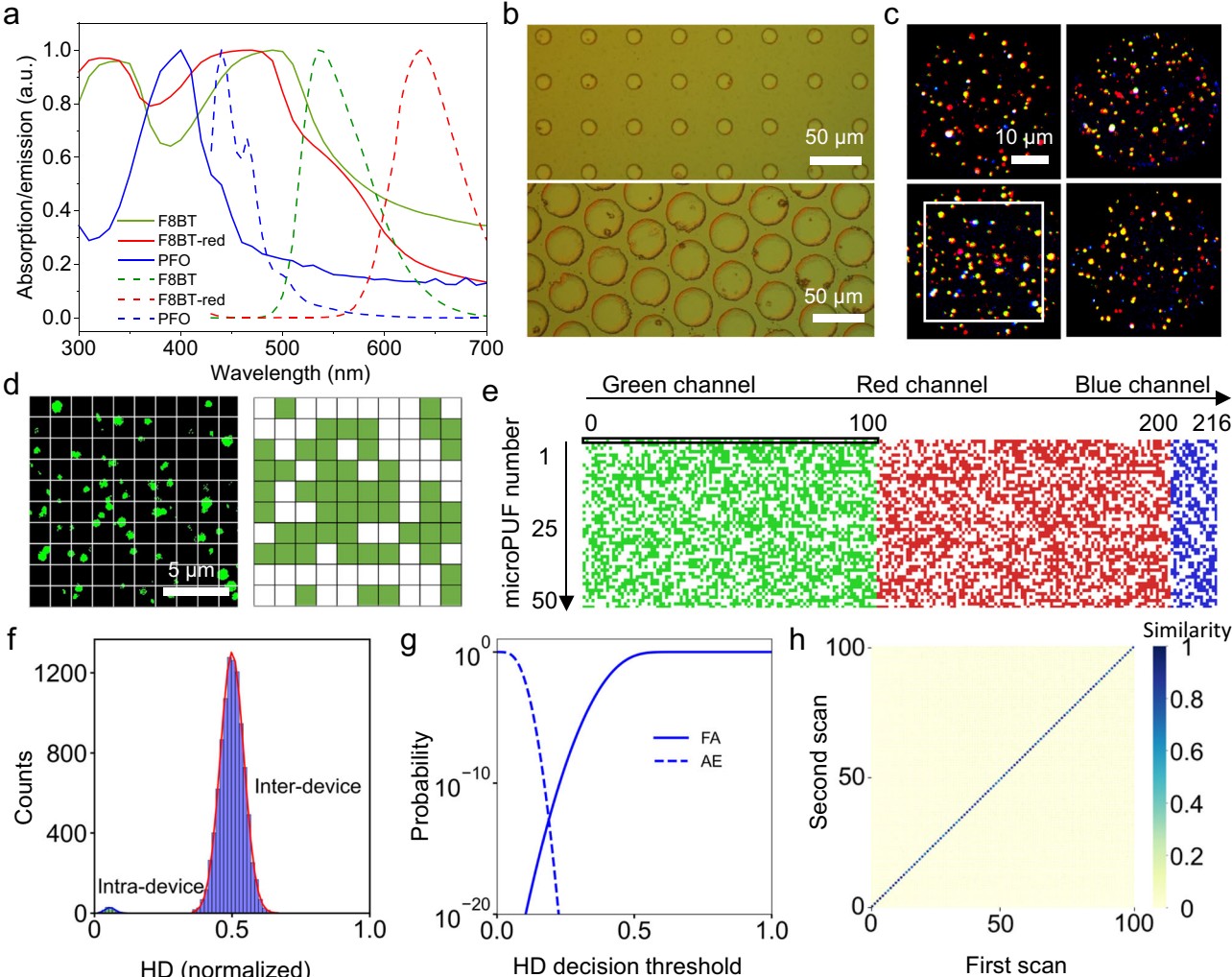

**Fig. 3 | Encoding process and characterizations of multichannel PUF devices.**
**a** Absorption (solid line) and emission (dotted line) spectra of F8BT, F8BT-red, and PFO NPs in ethanol solutions. **b** Optical image of the printed spot arrays with different spot sizes (20 μm diameter on the top and 50 μm diameter on the bottom). **c** Four representative fluorescence images from the spots embedded with a blend of F8BT, F8BT-red, and PFO NPs (Blue channel: $E_x$ 405 nm, $E_m$ 430–470 nm, 30% laser power, 150% gain. Green channel: $E_x$ 405 nm, $E_m$ 470–550 nm, 20% laser power, 100% gain. Red channel: $E_x$ 405 nm, $E_m$ 570–700 nm, 20% laser power, 100% gain). **d** Binarization of the area illustrated by the white square in **c**). **e** Digitized security keys with 10,800 bits are generated from 50 different microspots. **f** Inter- and intra-device Hamming distance of each PUF response. **g** Cumulative distribution functions, illustrating the probabilities authentication error (AE) and false authentication (FA) as a function of the decision threshold. **h** Heat map of intra-device (diagonal) and inter-device correlations obtained from 100 unique PUF patterns (generated using identical parameters). Source data are provided as a Source Data file.

excellent intra-correlations for the identical PUF under different challenges and the lack of any off-diagonal data points suggests that different PUFs are highly uncorrelated.

## Microspot fabrication to introduce multi-layered PUFs
A notable advantage of embedding the SPNs in a photoresist is that, in addition to the random distribution of the SPN aggregates, further stochastic processes could be explored through photolithography. Specifically, we could achieve microspots with fractal edges and random patterned arrays of microspots.

**Microspots with fractal edges.** Fractal structures are an attractive choice for PUFs as they generally exhibit highly random topography, statistical self-similarity and well-established analysis processes[38,39]. We therefore explored the generation of microspots with fractal edges as an additional PUF layer. We found that exposing positive photoresist (AZ 4562) to UV light (365 nm at 3 mW/cm$^2$) generated bubbles which grew with increasing exposure time, creating microstructured films

(Fig. 4a). However, these microstructures were not stable and could be washed away during development steps. To generate stable microstructures, we blended positive and negative photoresists to combine the bubble formation of the former with the cross-linking propensity of the latter. We started with a SU-8 2010:AZ 4562 (1:1.5) mixture, which formed unique patterns but required too high an exposure dose for the process (Fig. S9). This was most likely caused by the large thickness of the SU8 2010 films. Since SU8 2002 presented relatively lower viscosities and yielded thinner films after spin-coating (Fig. S10), we optimized blends of SU-8 2002:AZ 4562 (6:1, 5:1 and 4:1, v:v). These blends were patterned into microspot arrays with 120 s of UV exposure time (Fig S11a–d). Mixtures 5:1 and 4:1 yielded microspots with fractal edges (Fig. 4b). Lowering the UV exposure time (80 s) for the 5:1 mixture yielded microspots with a more exaggerated fractal-like structure (Fig. 4b) whereas mixture 4:1 was unstable to development under this lower UV exposure time[40–43].

We then analysed the complexity of the fractal pattern from microscopy images (Fig. S11b–e). The fractal dimension (D) quantifies

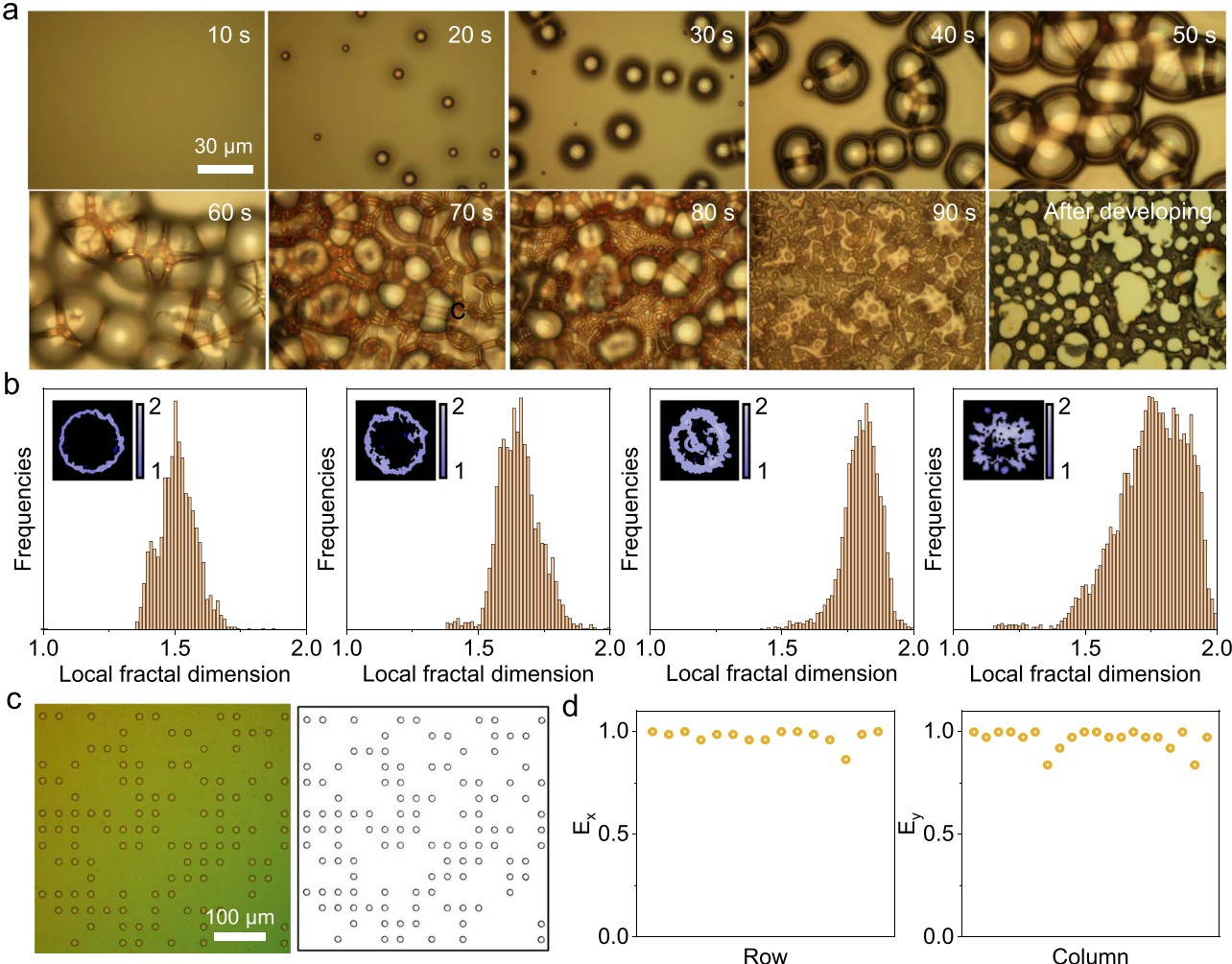

**Fig. 4 | Generation and characterization of stochastic processes across multiple length scales.** (a) Real-time observation of the microstructures formed by the decomposition of positive photoresist AZ 4562. The exposure process was performed by hard contact under 365 nm UV irradiation with an intensity of 3 mW/cm². b) Fractal structures with different complexity were generated by fine-tuning the lithography parameters. The ratios of SU 8 2002 to AZ 4562 and UV exposure time from left to right are 6:1, 120 s; 5:1, 120 s; 4:1, 120 s; 5:1, 80 s respectively. c) Stochastically distributed photoresist spot arrays can be binarized and encoded. d) The entropy along the x-axis and y-axis of the binarized bits, showing the randomness of the stochastic process. Source data are provided as a Source Data file.

the complexity of fractal patterns and is defined as:

$$D = \frac{\log N}{\log S} \qquad (4)$$

Where N is the number of miniature pieces, S is the scaling factor. Two-dimensional patterns typically present a fractal dimension in the range of 1–2 with a larger number indicating a higher complexity. We found by increasing the proportion of AZ 4562 and reducing the UV irradiation time, the microspots exhibited higher fractal dimensions (Fig. 4b). The resulting fractal structures could be precisely matched by a deep learning model, proving their capability as PUFs (Fig. S12). Therefore, by fabrication of fractal edges on the microspots we could achieve an additional layer of complexity in the microscale, significantly improving device robustness.

**Stochastic distribution of microspots.** Furthermore, we introduced an additional stochastic process to produce unique arrays of microspots. We found that microspot arrays could be partially delaminated by fine-tuning the thicknesses of photoresist films and developing conditions. We tested SU-8 2002, SU-8 2010 along with

different ratios of both to optimize the printing process. We observed a significant difference in film thickness as a function of these parameters (Fig. S10). The resulting thin films were then sonicated in developing solution to encourage delamination. With a ratio of 2:1 (v:v) between SU-8 2002 and SU-8 2010, we found the microspots to be partially removed after 1 min of ultrasonication. We then binarized the optical patterns of the resulting array to generate cryptographic keys (Fig. 4c).

The entropy of the remaining microspots was calculated for the rows and columns respectively (source code for entropy calculation can be found in Supplementary Materials). We found that both entropy values were mainly distributed around 1 (Fig. 4d), indicating the high randomness of the microspot distributions. In addition, we analysed the inter-correlation of five different microspot arrays, and found they presented high uniqueness values (Fig. S13). Even though the encoding capacities of the microspot array and fractal edges could not compete with that of the SPN fluorescent patterns, the readout is more straightforward and convenient. By combining them across multiple length scales, our PUF devices achieve both high security level with nanoscale details and simple identification by portable optical devices with macro-scale arrays.

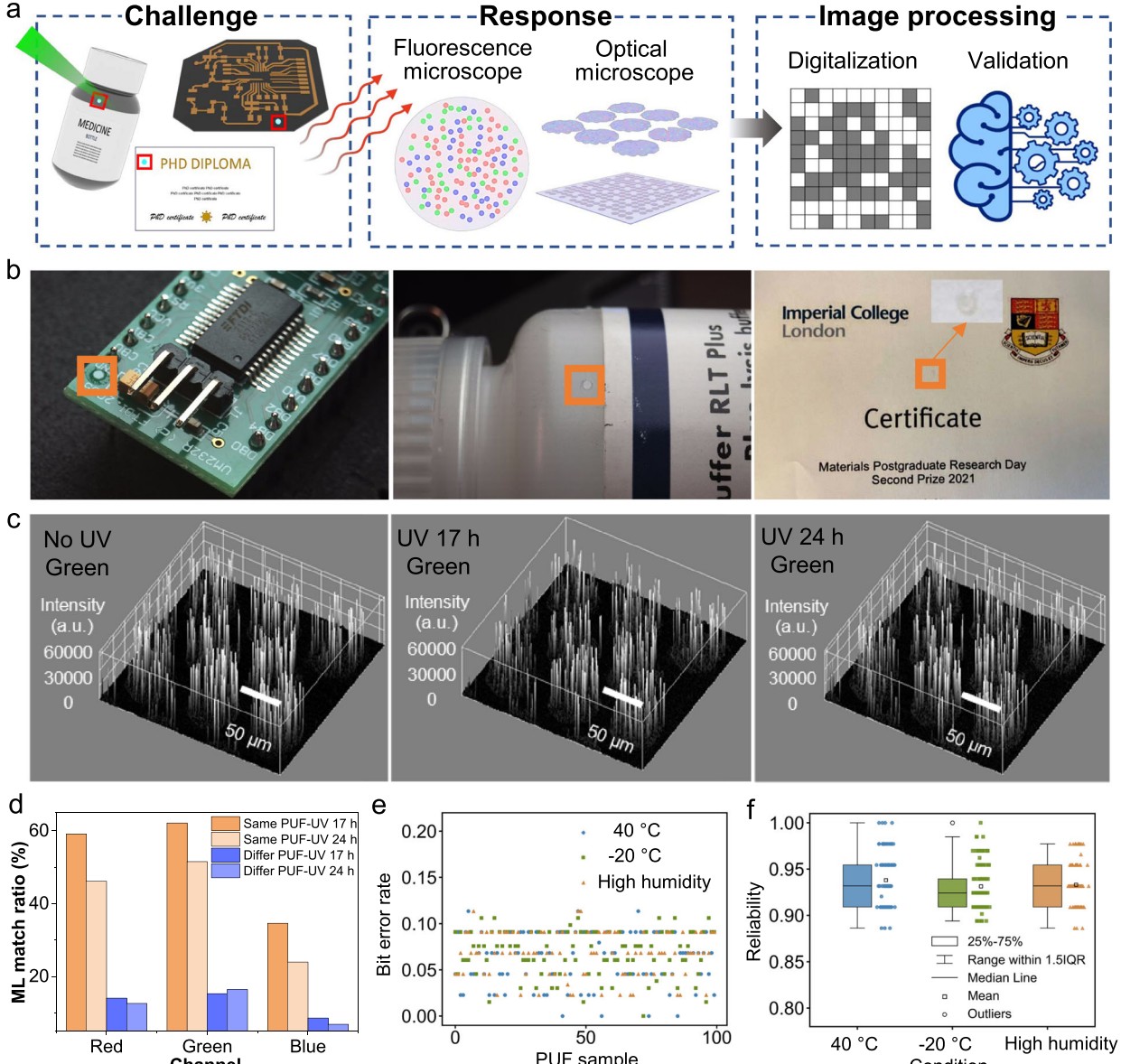

**Fig. 5 | Authentication process, practical application and stability test.** a) Workflow for the generation of challenge and response pairs, and machine learning assisted validation. Created in BioRender. Zhang, J. (2024) BioRender.com/i51y640. b) The optical PUF devices can be attached to different products including circuit boards, medicine packaging, and paper certificates for practical applications. c) Photostability test of the PUF devices. UV wavelength: 365 nm. UV intensity: 12 W/cm². Green channel: $E_x$ 405 nm, $E_m$ 470–550 nm, 20% laser power, 100% gain. d) Match ratios between UV irradiated PUF and the original PUF, based on the feature detecting results from the LoFTR algorithm. e) Bit error rate for each PUF under the storage conditions; f) reliability for each PUF in each storage condition. Values compared to images taken at day 0 and after one week. 100 PUFs were recorded per storage condition. Source data are provided as a Source Data file.

## Proof-of-concept application and stability testing

The encoding capacity of was calculated for the fluorescent PUFs by forming challenge-response pairs (CRPs, see Fig. 5a). Even with a small area (1 mm²), we found that an immense encoding capacity of ca. $10^{220322}$ can be achieved from the fluorescence channels alone. Table 1 shows a comparison of this value to the recent PUFs published in the literature (details of the calculations can be found in Supplementary Note 1, Supplementary Information). Practically speaking, it would be almost impossible for counterfeiters to mimic all pattern combinations. In addition, we could record, encode and identify both the optical images of the microspot arrays and fractal structures on the spot edges (Fig. S12). By storing the cryptographic keys in the data cloud and using a website for the encoding process, the end user could readily perform precise authentication with our device.

We then tested the compatibility of the PUFs on several appropriate substrates to demonstrate their implementation in a real-world setting. Devices were fabricated onto silicon wafers, plastic sheets, aluminium foil and paper (Fig. S14). As a proof-of-concept, we punched spot arrays on a plastic sheet into rounded tablets with a diameter of 1 mm. To assess and showcase the performance of our tablets in various real-world scenarios, we glued them to a range of material surfaces that they would likely find application on (Fig. 5b).

We chose SPNs in part due to their reported superior photostability. Therefore, we evaluated the photostability of the optical PUF devices under prolonged UV irradiation (24 h of exposure being equivalent to ca. 640 h of solar UV irradiation[44]). After exposure for 17 and 24 h, no significant decrease in fluorescence intensity was observed (Fig. 5c & Fig. S15), suggesting that these PUF devices should exhibit excellent long-term stability under natural light exposure. PUF

**Table 1 | Encoding capacity of SPN-PUF compared to recently reported PUFs**

| Type of PUF | Materials | Bits | Number of CRPs | Size | Reference |
|---|---|---|---|---|---|
| Electrical | Metal oxide thin film | 28-bit | Not mentioned | $20 \times 20$ mm$^2$ | [3] |
| Concealable memristor | Hafnium oxide | 128-bit | Not mentioned | $427 \times 352$ μm$^2$ | [4] |
| Electro-magnetic | CPAL devices | 256-bit | $2^{289}$ | Not mentioned | [5] |
| Optical | Quantum dots | Not mentioned | $4.7 \times 10^{202}$ | $\sim 80 \times 80$ μm$^2$ | [12] |
| Optical | Carbon dots | Not mentioned | $10^{63,593}$ | $700 \times 700$ μm$^2$ | [17] |
| Optical | Gold | Not mentioned | $10^{348}$ | $100 \times 100$ μm$^2$ | [39] |
| Optical | Silk fibres | Not mentioned | $2^{768}$ | $1 \times 1$ cm$^2$ | [40] |
| Optical | Planar spin glass | Not mentioned | $10^{55}$ | $600 \times 600$ μm$^2$ | [41] |
| Optical | Organic Crystal | Not mentioned | $10^{7000}$ | $1 \times 1$ mm$^2$ | [42] |
| Optical | Lanthanide luminescence | Not mentioned | $6 \times 10^{104}$ | $30 \times 30$ μm$^2$ | [43] |
| **Optical** | **SPNs** | **731471** | $\mathbf{2^{731471} \approx 10^{220322}}$ | **$1 \times 1$ mm$^2$** | **This work** |

images were also compared by the LoFTR deep learning model, before and after UV exposure. Figure S16 shows the feature matching process using the model using one PUF pair as an example, the remaining pairs can be found in Fig S17-19. To quantitively analyse the identification process after UV irradiation, we considered the match numbers from the same image as the maximum number of matches that the algorithm can possibly achieve, and these are used to normalize other results to generate match ratios. Match weights are factors which indicate the reliability of the matches. In all RGB channels, we found the match ratios between the same PUF before and after UV irradiation to be much higher than those between different PUFs, demonstrating the high stability and reliability of the PUF devices (Fig. 5d).

Aging tests were then conducted on the multichannel devices. SPNs embedded in photoresists were sustained at an elevated temperature (40 °C), low temperature (−20 °C) and high humidity (85%) for one week. Images were then compared to those taken initially to measure the bit error rate. Figure 5e shows the bit error rate calculation and values obtained for 100 PUFs in each device. In all cases the PUFs show reliabilities above 90% (Fig. 5f) suggesting the excellent performance of the devices under prolonged exposure to these conditions. Details of calculations can be found in Supplementary Note 2, Supplementary Information.

CLSM was employed throughout this work to illustrate the use of SPN taggants. We wanted to investigate the use of a smaller, portable and lower cost fluorescent reader which would be more accessible. Here, we implemented a USB green-fluorescent microscope (Dino-Lite Digital Microscope, ca. £500) to image and authenticate the PUFs. Figure S20a–c shows fluorescent images taken with the portable microscope, in which SPN aggregates can clearly be seen. Employing the same LoFTR algorithm as described above showed excellent matching scores between identical images (Fig. S20d) and images with varied exposure time (Figure S20e). Furthermore, low matching scores were found for images in two different PUF regions (Fig. S20f). An image of the microscope can also be found in Figure S20g. This illustrates that the authentication of the fluorescent PUFs could be achieved with a portable microscope, making these PUFs practical for real-world product authentication.

We successfully created optical PUF devices using stochastic methods across multiple length scales, achieved primarily through material and process engineering. We found that SPNs exhibited high brightness and photostability in comparison to other commonly used fluorescent materials. Through exploration of surfactants and anti-solvents, we discovered that small SPN aggregates could be formed and embedded in photoresist thin-films. Spin-coating was used to produce thin-films here but in future we envisage that more scalable techniques such as roll-to-roll, slot-die or dip-coating could be readily employed. The obtained particles can be easily detected by CLSM with relatively low laser power, which aided the long-term photostability

required for PUF devices. We then randomly dispersed red, green and blue SPNs in photoresists which were localized in specific areas after cross-linking. As a result, SPNs could be dispersed in microspot arrays with different spot sizes. Each microspot offers 216-bit strings after the encoding process, achieving 10,800-bit cryptographic keys in a similar area of a single mammalian cell. Besides the high encoding capacities, the SPN distribution also showed near-ideal bit uniformity, uniqueness, and readout reproducibility. We also demonstrated two extra stochastic processes, which were independent of the SPN distributions, through the design of photoresist recipes and investigation of developing conditions. This was, in part, made possible due to the high photostability of the SPNs which could withstand the intense UV irradiation. This stochastic fabrication yielded PUF devices with significant complexity. A LoFTR deep learning model was modified and applied for PUF identification, and exhibited excellent performance. We then explored the scope of the potential substrates that were compatible with the fabrication processes. A microspot array embedded with SPNs could be printed on a silica wafer, glass slide, plastic plate, and paper. Finally, we affixed the PUF devices to different surfaces (circuit boards, medicine packaging, and paper certificates) showcasing their potential for practical, real-world applications. The high substrate tolerance of these PUF devices, coupled with the high stability and information density, could make our approach particularly attractive for high value products where either the esthetics are critical (e.g. luxury watches) or space is of a premium (e.g. semiconductors). Our approach synergizes independent stochastic processes with SPNs as superior functional taggants, which paves the way for the next generation of PUF devices and offers a valuable perspective for the platform design in other related fields (e.g. information encryption and storage).

## Methods
### Synthesis of F8BT-red
The polymerisation was carried out using conventional Suzuki-Miyaura coupling chemistry. To a 20 mL high pressure microwave vial was added 4,7-dibromo-5-fluoro-2,1,3-benzothiadiazole (0.6 mmol), the 4,7-*bis*(5-bromothiophen-2-yl)-5-fluoro-2,1,3-benzothiadiazole, 9,9-dioctyl-9H-fluorene-2,7-diboronic acid *bis*(pinacol) ester (0.6 mmol), Pd(PPh$_3$)$_4$ (2 mol %), and a magnetic stirrer bar. This was sealed with a microwave vial cap and flushed with argon following which degassed toluene (5 mL) and an aqueous solution of 2 M Na$_2$CO$_3$ (0.6 mL) were added. The mixture was heated to 120 °C and stirred for 3 days. The solution was then cooled to room temperature and precipitated into MeOH (100 mL). The precipitate was filtered into a cellulose thimble and washed using a Soxhlet apparatus under Argon with MeOH, acetone, hexane and finally chloroform. The chloroform was concentrated to less than 10 mL, precipitated into MeOH (100 mL) and filtered yielding the final product. Product isolated as a red solid (366 mg, 100 %). $^1$H NMR (400 MHz, CDCl$_3$) δ 8.29–7.54 (m, 7.5H), 1.29–0.67 (m, 34H). $^{19}$F NMR (400 MHz,

CDCl$_3$) δ -114 (s, 1 F). Mn = 82.9 kDa, Mw = 203.6 kDa, Mw/Mn (Đ) = 2.5. See Fig. S5 for scheme and NMR spectra.

## Preparation and characterization of SPNs

F8BT (M$_n$ ≤ 25000 Da, Sigma), PFO (M$_n$ = 29167 Da, Ossila) and the surfactant PS-$g$-FEO (M$_n$ = 6000-$g$-1400, Polymer Source. Inc.) were purchased and used as received. F8BT solution (1 mg/mL in THF) and PS-$g$-FEO solution (10 mg/mL in THF) were prepared. For size optimization, 10, 30 or 50 µL PS-g-FEO were mixed with 50 µL F8BT solution, respectively. Then, 50 µL mixture was added to 250, 500 or 1000 µL milli-Q water. The obtained solutions are placed in a fumehood overnight to evaporate THF and yield nanoparticles. The size distribution of the SPNs was performed by a Wyatt DynaPro III plate reader. 30 µL original or diluted solutions was added into a 384-well plate. Each formulation was repeated three times and each repeat was measured three times.

## Photolithography process

The SUSS MicroTec MJB3 photolithography was used to print the spot array. Negative photoresists SU-8 2002, SU-8 2010 (Kayaku Advanced Materials, Inc.) and positive photoresist AZ 4562 (Imprint Micro-chemicals GmbH) were purchased and used as received. For a standard process, 50 µL SPN solution was added into 500 µL SU-8 2002. The mixture was treated by a short ultrasonication (20 s, default process 7), then spin-coated (2000 rpm, 30 s, SUSS MicroTec LabSpin platform) into a glass substrate. The obtained film was placed on a hotplate for soft bake (75 °C, 60 s). The following exposure process was performed by hard contact under 365 nm UV irradiation with an intensity of 4.7 mW/cm$^2$ for 60 s. It is noted that the intensity of UV irradiation increased from 3 to 4.7 mW/cm$^2$ after realignment and calibration by lithography engineers. Afterward, a post-exposure bake took place directly at 110 °C for 60 s. The glass slide was cooled down to room temperature and placed in a chamber filled with SU-8 developer for 60 min with shaking. Then, the glass slide was washed with acetone, isopropanol, and acetone again before drying with compressed air.

SU-8 2002 and SU-8 2010 mixture with ratios at 1:0.5, 1:1, and 1:2 (v:v) were used for the thickness optimization. The step height measurements were performed by an Alpha-Step D-500 stylus profiler. SU-8 2002 and AZ 4562 mixtures with ratios at 6:1, 5:1, and 4:1 (v:v) were employed for the generation of the fractal edge structure. FracLac, a plugin of ImageJ was employed for fractal analysis. The 2D fractal dimension map was generated using active scan mode with default settings (box counting binary, no filters). The local fractal dimension distribution was obtained by the 'subscan' option, and the content style was chosen as a separate bar.

## Fluorescence imaging and photostability

All the fluorescence images were obtained by a confocal microscope (Leica Stellaris 5 Light Sheet). For photobleaching, all the fluorescent materials were distributed in polymer films for measurement. Specifically, the taggant solutions were prepared at a concentration of 0.1 mg/mL, and then 30 µL of each solution was added into 350 µL PVA water or PS dichloromethane solution (60 mg/mL). After short vortexing, the mixture was spin-coated on a thin glass slide at the speed of 2000 rpm for 30 s to evaporate the solvents. All samples were excited at the same wavelength of 405 nm and observed under a 63X oil objective. Before the photobleaching measurement, the area of interest was focused and selected with a laser power of 15% and a gain value of 150%. Except for carbon dots, a laser power of 15% and a gain value of 500% were applied to get a decent fluorescence signal. Afterward, the laser power was increased to 80% and the gain value was reduced to 80 to avoid saturation. Images were recorded every 15 s for 15 min (1024 × 1024 resolution). Image analysis was performed using the Fiji distribution of ImageJ[45]. To mitigate cosmic noise, the images were processed using a moving average filter. This involved replacing each pixel value with the median of the surrounding 1 × 1 pixel window, including the pixel itself. The size of a single pixel is 0.167 µm x 0.167 µm. The nanoparticle population within each image was differentiated from the background using a thresholding technique (Triangle method). Subsequently, the total intensity of the nano-particle population was quantified. For the photobleaching analysis, the total intensity of images captured over the course of the photo-bleaching process was normalized to the initial signal intensity at the starting time point. This normalized intensity was then plotted as a function of laser exposure time. The particles were extracted from the images and their mean intensities were plotted as a function of laser time in Fiji.

For UV stability measurements, the printed SPN microspots were treated under irradiation (305 nm, 10 mW/cm$^2$) for 6 h. Fluorescence images were recorded before the treatment, after 3 h, and after 6 h by confocal microscopy (Blue channel: E$_x$ 405 nm, E$_m$ 430–510 nm, 100% laser power, 250% gain. Green channel: E$_x$ 430 nm, E$_m$ 500–550 nm, 30% laser power, 150% gain. Red channel: E$_x$ 430 nm, E$_m$ 650–700 nm, 30% laser power, 150% gain.). Their 3D intensity maps were obtained by Fiji.

## Generation of the PUF keys

The raw fluorescent images were normalized and converted to a binary bitmap[32] (Fig. 3d). We then use a binning process to remove 50 pixels from the edges of microspots. To ensure high reproducibility, we use 10 × 10 pixels as a unit for binarization instead of just a single pixel (for F8BT and F8BT-Red) and 50 × 50 pixels for PFO SPNs. The intensity of each unit was calculated and upon exceeding the median, the unit was assigned to 1-bit. We then combined three challenge-response pairs together with the final digitized key resulted in 216 bits per microspot, with 100 bits from the red and green channels respectively and 16 bits from the blue channel.

## SPN-polymer film preparation and stability in artificial sweat

SPN-PVA film: 30 mg PVA (Mw = 13–25 kDa) and 500 µL DI water were added into a 1.5 mL EP tube. The EP tube was treated in a 90 °C water bath for several minutes until PVA was dissolved. After the tube was cooled down to room temperature, 50 µL SPN solution was added and vortexed for a homogenous mixture. The solution was then spin-coated at 2000 rpm for 30 s onto a glass slide to generate a SPN-PVA film.

SPN-PS film: 30 mg PS (Mw = 192 kDa) was dissolved into 500 µL DCM. A 50 µL aliquot of SPN solution was added and vortexed. The obtained turbid mixture was directly spin-coated at 2000 rpm for 30 s onto a glass slide to generate a SPN-PS film.

The fluorescence images of both films were collected by a con-focal microscope (Leica SP8 - STELLARIS 5 Inverted Light Sheet). A 63x/1.40 oil objective was applied. Images were captured under the excitation of a 405 nm laser with a laser power of 10% and 20 gain value. Afterward, the films were treated in a Petri dish filled with 20 mL water solution with 0.9 g/L NaCl and 0.2 g/L KCl for 30 min. The two glass slides were washed with DI water and dried with compressed air before their fluorescent signal was recorded again by the confocal micro-scope with the same parameters.

## Data availability

Data supporting the findings of this study are available within the article and its supplementary files. Source data are provided with this paper and additional raw data are available online at DOI: 10.5281/zenodo.14364817. Source data are provided with this paper.

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

## Acknowledgements

J.Z., J.P.W. and M.M.S. acknowledge support from the grant from the UK Regenerative Medicine Platform "Acellular / Smart Materials – 3D Architecture" (MR/R015651/1). A.C. and M.M.S. acknowledge support from CRUK early detection and diagnosis primer award (Grant No.100063) and EPSRC IRC Agile Early Warning Sensing Systems for

Infectious Diseases and Antimicrobial Resistance (Grant No. EP/R00529X/1). L.S. and M.M.S. acknowledges the Engineering and Physical Sciences Research Council (EPSRC – EP/L016702/1 – through the Plastic Electronics Center for Doctoral Training) and the Rosetrees Trust. K.X. acknowledges support from the Engineering and Physical Sciences Research Council through a UKRI Postdoc Guarantee Fellowship (EP/X027287/1). We acknowledge technical support of the Facility for Imaging by Light Microscopy (FILM) at Imperial College London. FILM is part-supported by funding from the Wellcome Trust (grant 104931/Z/14/Z), BBSRC (grant BB/L015129/1 and BB/T017929/1). We acknowledge technical support for photolithography from the CBIT Clean Room Core facility in Imperial College London. For the purpose of open access, the authors have applied a Creative Commons Attribution (CC BY) licence to any Author Accepted Manuscript version arising. We are grateful to Prof. Martin Heeney and Dr. Martina Rimmele for the GPC characterization.

## Author contributions

J.Z. and A.C. contributed equally to this work. J.Z. and M.M.S. conceived the general idea. J.Z., A.C., and J.P.W. designed the project. J.Z. performed the majority of the experiments, including SPN synthesis and characterization, photolithography process, and PUF characterization. A.C. contributed to polymer synthesis and characterization. K. X. contributed to the application of PUF labels on different products and assisted the photolithography process. J.T. performed the picture processing for photobleaching curves. L.S. contributed to polymer synthesis. J.Z. and A.C. prepared the manuscript, while K.X., J.P.W., J.T., L.S., and M.M.S. all revized the paper. M.M.S. supervised the study.

## Competing interests

A patent, describing the technology herein, has been filed under UK Patent Application Number 2408586.2 by Oxford University Innovation Limited and Imperial College Innovations Limited with JZ, AC and MMS as named inventors. The remaining authors declare no competing interests.
