## [Peer Review File · Nature Communications]

REVIEWER COMMENTS

Reviewer #1 (Remarks to the Author):

In this article, the authors proposed SPNs-based PUF labels with optimized fabrication process. Compared to existing SPN-based PUF labels, their labels exhibit two advantages: repeated readout and independent stochastic processes. This article presents two crucial advantages. First, the well-organized manuscript ensures reader-friendly comprehension. Second, it provides robust scientific evidence supporting the outstanding performance of their PUF label. The performances include brightness, stochastic particle distribution, multiple length scales, and photostability. However, they haven't identified any outstanding innovation or breakthrough. Specifically, while the authors demonstrate optimized performance of their SPN-based PUF labels, the PUF label lacks a unique property or a matched application scenario. Before considering for publication in NC, the authors should address the following questions:

Q1. Compared to the previous fabrication process of SPN-based PUF label, whether authors make some crucial change causing the optimization of label performance?

Q2. Brightness, stochastic particle distribution, and photostability have been proved for previous PUF labels (quite a few examples existing in literature). Whether authors can point out a unique property of their PUF label and stress it in the manuscript?

Q3. The authors can explore specific application scenarios where their PUF label's outstanding performance aligns with real-world needs.

Q4. Authors can describe their main innovation or breakthrough from their own perspectives.

Reviewer #2 (Remarks to the Author):

This manuscript presents an innovative approach to addressing the critical issue of counterfeiting by developing physical unclonable functions (PUFs) with enhanced encoding capabilities and resilience. The use of semiconducting polymer nanoparticles (SPNs) as fluorescent taggants is highly innovative, offering significant improvements in brightness, photostability, and size tunability over current taggants. The multi-scale design of the PUFs, incorporating nanoscale, microscale, and macroscale features, enhances their complexity and security. The experimental results are robust, showing high resilience to UV radiation and ease of detection. The integration of deep-learning models to enhance performance and accessibility is particularly impressive, making

complex concepts accessible to a broad audience. The potential applications of this technology in next-generation security devices are vast. The authors effectively highlight the practical implications of their work for general end users. The strategy presented for developing PUFs with multiscale features could set a new standard in the field, paving the way for further research and development.

I have a few comments that should be addressed before publication:

For optical PUFs, the challenge-response-pairs (CPRs) do not appear to be very high. It would be beneficial to show more detailed distinctions based on incident angle or wavelength. Additionally, providing a table comparing these CPRs with those reported in recent papers would significantly highlight the manuscript. It appears that some of the latest papers are missing from the citations.

To demonstrate practical applicability, it is necessary to present reliability tests, including temperature, humidity, and aging tests. Observing how key characteristics like the Bit Error Rate (BER) change is crucial.

The machine learning results seem to disrupt the overall narrative of the manuscript. It would be better to move these results to the supplementary information. The figureset is also somewhat difficult to understand and could be improved for clarity.

While demonstrating scalability might be challenging for an initial study on PUFs, it would be beneficial to discuss the potential for scalability to some extent.

The process of measuring and obtaining the PUF key is somewhat omitted. Providing more detailed information would be helpful for the readers.

Reviewer #3 (Remarks to the Author):

The authors have fabricated PUFs by using semiconducting polymer nanoparticles (SPNs) as fluorescent taggants. The PUFs consist of designs from nanoscale to macroscale. With the assistance of a deep-learning model the authors show that the PUFs have both near-ideal performance and accessibility for general end users, offering a strategy for next-generation security

devices. This work may help advance the development of optical PUFs, which is of great current interest. I would like to ask the authors to consider the following points.

1. The authors begin with the screening of fluorescent nanoparticles. This is in fact quite challenging given the large quantity of fluorescent nanoparticles. Only 4 types of fluorescent nanoparticles are compared in the current work. Other typical fluorescent nanoparticles such as recently reported silicon nanoparticles (Nature Communications 15, 3203 (2024). DOI10.1038/s41467-024-47479-y) are missing. Please consider a better strategy for the comparison (e. g., combining own experiments with published results in literature).

2. Could the authors more clearly justify the choice of F8BT NPs among all SPNs?

3. The optical signal is collected by using confocal laser scanning microscopy. Please comment on the practical use of this technique in terms of convenience, portability, etc..

4. Please tell the exact size of a pixel in the manuscript.

We thank the reviewers for their thoughtful comments and suggestions for improving this manuscript. Please see below for our responses to each comment individually. For clarity, our responses are in blue and the manuscript changes are in red. In addition, the resulting manuscript and supporting information changes are highlighted in yellow.

Reviewer 1

In this article, the authors proposed SPNs-based PUF labels with optimized fabrication process. Compared to existing SPN-based PUF labels, their labels exhibit two advantages: repeated readout and independent stochastic processes. This article presents two crucial advantages. First, the well-organized manuscript ensures reader-friendly comprehension. Second, it provides robust scientific evidence supporting the outstanding performance of their PUF label. The performances include brightness, stochastic particle distribution, multiple length scales, and photostability. However, they haven't identified any outstanding innovation or breakthrough. Specifically, while the authors demonstrate optimized performance of their SPN-based PUF labels, the PUF label lacks a unique property or a matched application scenario. Before considering for publication in NC, the authors should address the following questions:

We appreciate the kind comments about our work and the constructive feedback. We have addressed each comment in full below.

Q1. Compared to the previous fabrication process of SPN-based PUF label, whether authors make some crucial change causing the optimization of label performance?

We are sorry for the confusion here, which is most likely as a result of the introduction section. There are no previously reported SPN-based PUFs, to the best of our knowledge this is the first.

The previous anti-counterfeiting studies we discussed in the introduction are SPN-based, but not PUFs. We prepared a table to show the differences between our work and SPN-related literature (Table. R1-1). All these literature examples illustrate the use of SPNs as anti-counterfeiting inks, proving the concept. Since the macro-patterns are fabricated by determinative approaches, these security tags are vulnerable to skilled counterfeiters. They are classified as cloneable tags, even though their inks can be very sophisticated. In our case, we showcase a practically unclonable device by implementing stochastic processes, image digitalization, cloud datasets, reliability testing and smart identifications. Furthermore, our PUF devices have improved performance when comparing to PUF devices based on other fluorescent taggants (quantum dots, carbon dots, and perovskite NPs, please see Figure 2 in the manuscript). Besides the excellent taggant properties, the structure of our PUF devices are designed with independent stochastic processes covering multiple length-scales (including nanoparticles, microstructures, and macro-arrays). Depending on the specific applications, the devices allow free combinations of the patterns from different scales. Therefore, the device is not only highly robust but also flexible enough to satisfy different scenarios.

Table R1-1. Comparison between SPN-anticounterfeiting related literatures and this work.

Application	Strategy	Approach	Pattern	Safety	Smart identification	Literature
Ink	Full-Color fluorescence	Inkjet printing	Macro-pattern	Cloneable	Not involved	Small 10 , 4270-4275 (2014)
Ink	Responsive luminescence	Hand-written	Macro-pattern	Cloneable	Not involved	ACS Appl. Mater. Interfaces. , 10 , 39214-39221 (2018)
Ink	Responsive luminescence	Inkjet printing	Macro-pattern	Cloneable	Not involved	ACS Appl. Mater. Interfaces. , 9 , 30918-30924 (2017)
Ink	Multi-responsive luminescence	Hand-written	Macro-pattern	Cloneable	Not involved	Sci. Adv. 8 , eadd1980 (2022)
Ink	Responsive luminescence	Hand-written	Macro-pattern	Cloneable	Not involved	Chem. Commun. , 59 , 2469-2472 (2023)
PUF device	Solvent engineering	Lithography	Nano-/micro- and macro-pattern	Unclonable	Yes	This work

We have changed the description in the introduction to more accurately describe the previous work in this space and added the citations from the table above to the manuscript.

“Recent works have illustrated semiconducting polymers in anti-counterfeiting devices, wherein polymer inks were printed in unique patterns²¹ and engineered to show reversible color changes in response to chemical and physical stimuli²²⁻²⁴. Similar approaches have also been shown with polymer-photoswitch conjugates in both paper and gels^{25,26}. However, this photoswitch approach is susceptible to degradation upon repeated stimulation, especially when the process involves harsh conditions¹². Furthermore, counterfeiters could copy the device by synthesizing the same underlying photo-active materials once they are discovered²⁷. SPN-based PUF devices have yet to be explored and have the potential to outperform current fluorescent optical PUFs. “

Q2. Brightness, stochastic particle distribution, and photostability have been proved for previous PUF labels (quite a few examples existing in literature). Whether authors can point out a unique property of their PUF label and stress it in the manuscript?

We agree with the sentiments here, there are other optical PUFs based on fluorescent materials in the literature which we have cited in the introduction. However, in this work (Figure 2 in our manuscript) we illustrate that under the same measurement conditions SPNs are brighter and more photostable than the reported competition (quantum dots, perovskites and carbon dots). Also, due to the large Stokes shift, our SPN taggants can achieve RGB emission under a single excitation wavelength in the visible region. This is beneficial for practical applications which is otherwise challenging for other fluorescent materials.

The enhanced photostability of the SPNs permitted the use of photolithography techniques, which is typically limited to non-fluorescent materials due to the involvement of UV irradiation. Therefore, the creation of the multiple independent stochastic processes was possible, while literature typically only contains a single stochastic process. In this work we illustrate this combination of fluorescent SPNs with random microspot arrays and fractal edging (detailed in Figure 3 and 4, respectively). The development of SPN probes as new and powerful PUF taggants, the structural design of the PUF devices and the advanced PUF properties are the unique properties of our PUF.

To illustrate how this work compares to published PUFs in the literature, we have compiled the following table:

Table R1-2. Comparison between PUF literatures with other optical taggants and this work.

Taggant	Stochastic process	Performance			Reference
		Uniqueness (%)	Encoding Capacity	AI-based identification	
Metallic nanopatterns	Single	45.39 (2-bit)	2^{10000}	No	Nat. Electron. , 5 , 433–442 (2022)
Au networks	Single	~30-43 (2-bit)	$\sim 2^{13920}$	Yes	Nat. Commun. , 14 , 2185 (2023)
Graphene	Single	47 (2-bit)	2^{64}	No	Nat. Electron. , 4 , 364–374 (2021)
Metasurfaces	Single	46 (2-bit)	2^{289}	Yes	Sci. Adv. 9 , eadg7481(2023)
Planar spin glass	Single	50.2 (2-bit)	$2^{184.3}$	Yes	Adv. Mater. 35 , 2303077 (2023)
Carbon dots	Single	49.8 (2-bit)	$2^{211,250}$	Yes	Nat. Nanotechnol. , 18 , 1027–1035 (2023)
Fe ₂ O ₃ /C	Single	93.7 (16-bit)	2^{448}	Yes	Nat Commun. 15 , 1040 (2024)
Semiconducting polymer dots	Ternary	50.0 (2-bit)	$\sim 2^{640,000}$	Yes	This work

Noncomparable

Low

High

To further clarify these advantages to the reader, we have included additional discussion in the manuscript.

“To overcome this challenge, we introduced a UV cross-linking step to ‘lock-in’ the SPN aggregates by using a photoresist (SU-8) as the polymer matrix for spin-coating. The high photostability of the SPNs (Figure 2d) meant that photobleaching was not of concern during UV exposure.”

“We also demonstrated two extra stochastic processes, which were independent of the SPN distributions, through the design of photoresist recipes and investigation of developing conditions. This was, in part, made possible due to the excellent photostability of the SPNs which could withstand the intense UV irradiation.”

A variation of the table above is also now included in the supplementary information (Supplementary Note 1), as a response to reviewer 2 (see below).

Q3. The authors can explore specific application scenarios where their PUF label's outstanding performance aligns with real-world needs.

We highlighted potential real-world applications in our conclusion but we realise that we did not stress specific applications where our technology would potentially excel. We thank the reviewer for highlighting this important point.

As stated in the manuscript, the information stored in our PUF is very dense, with 10,800-bit cryptographic keys stored in the size of a mammalian cell. This high density could lend itself to high-value items such as designer watches and jewellery – where authenticity is of large concern but anti-counterfeiting tags cannot affect the aesthetics of the product.

Another application is bank notes; the Euro, for example, has >10 individual anticounterfeiting measures which are sophisticated but all classified as cloneable tags (see Figure 1 in *Nat. Rev. Chem.*, 2017, 1, 0031). These could be potentially replaced by a single PUF tag from our approaches.

Security tags for semiconductor chips are also an important industry issue. Silicon-PUFs are a popular choice here but as discussed in the introduction, they suffer from low reliability. Our devices could be a suitable replacement as the photolithography process in our approach is compatible and widely used in the semiconductor industry. There is very little available space on a semiconducting chip, so the high density is also an advantage here. We have proved the concept that our PUF device can be applied on circuit boards in Figure 5 in the manuscript.

Furthermore, in response to Reviewer 3, we have now added an additional figure (S20, see below) where we show how our fluorescent PUF can be read with a small hand-held microscope, which highlights the expensive instruments are not required for readout. We showcase an affordable commercial-grade USB fluorescent microscope (Dino-Lite, ~£500) to image the fluorescent PUF. The photos were taken on a black surface to minimize reflection. We have tested the same process flow using images with different exposure times and regions (in Supplementary Figure 20 a and b below). Analysis of two identical images (same region and same exposure) shows a high number of matching regions (4256). Benchmarking with this result, analysis among two different exposure times shows the same result, indicating the robust analysis pipeline to level the discrepancy in brightness. Expectedly, analysis shows only negligible matching regions from images taken in different regions. These results verified that we can also use affordable, portable, low-cost USB fluorescent microscopes for authentication of our PUF.

Supplementary Figure 20. Images recorded by a USB microscope and the following feature matching analysis with LoFTR algorithm. Patterns imaged in a) long exposure time (1 s) in area 1, b) short exposure time (1/4s) also in area 1, c) long exposure time (1 s) in area 2. (d) Feature matching between the two images obtained from same exposure time and area. e) Feature matching between the two images obtained from same PUF but different exposure times. f) Feature matching between the two images obtained from different PUFs recorded with same exposure time. g) Photograph of USB microscope with a ten pence coin for size comparison.

We have changed the text to discuss this in more depth.

“Finally, we affixed the PUF devices to different surfaces (circuit boards, medicine packaging, and paper certificates) showcasing their potential for practical, real-world applications. The high substrate tolerance of these PUF devices, coupled with the high stability and information density, could make our

approach particularly attractive for high value products where either the aesthetics are critical (e.g. luxury watches) or space is of a premium (e.g. semiconductors). Our approach synergizes independent stochastic processes with SPNs as superior functional taggants, which paves the way for the next generation of PUF devices and offers a valuable perspective for the platform design in other related fields (e.g. information encryption and storage).”

“CLSM was employed throughout this work to illustrate the use of SPN taggants. We wanted to investigate the use of a smaller, portable and lower cost fluorescent reader which would be more accessible. Here, we implemented a USB green-fluorescent microscope (Dino-Lite Digital Microscope, ca. £500) to image and authenticate the PUFs. Figure S20a-c shows fluorescent images taken with the portable microscope, in which SPN aggregates can clearly be seen. Employing the same LoFTR algorithm as described above showed excellent matching scores between identical images (Figure S20d) and images with varied exposure time (Figure S20e). Furthermore, low matching scores were found for images in two different PUF regions (Figure S20f). An image of the microscope can also be found in Figure S20g. This illustrates that the authentication of the fluorescent PUFs could be achieved with a portable microscope, making these PUFs practical for real-world product authentication.”

Q4. Authors can describe their main innovation or breakthrough from their own perspectives.

The key breakthrough in this work is the generation of an optical device which consists of three complementary PUF layers, fabricated by multiple independent stochastic techniques. This builds in complexity from a random array of microspots, to each spot having a unique fractal edge and finally a unique array of SPNs within each microspot. The strategy yields a practically unbreakable device when used in tandem.

Furthermore, depending on the level of security required for the end application, this technique gives the flexibility to fabricate devices consisting of only a microspot array, with or without fractal edging.

There are also more innovations across this work:

- Size manipulation of nanoparticles via solvent engineering to achieve larger nanoparticles (>200 nm) for easier readout.
- Long-term stability of SPN-devices under UV-irradiation, a sweat mimic and different temperatures.
- By polymer design, we achieved RGB emissions from single wavelength excitation, effectively simplifying image acquisition in practical applications.

We feel with the previous answers we have stressed the above points in the manuscript so have not made any further edits.

Reviewer 2

This manuscript presents an innovative approach to addressing the critical issue of counterfeiting by developing physical unclonable functions (PUFs) with enhanced encoding capabilities and resilience. The use of semiconducting polymer nanoparticles (SPNs) as fluorescent taggants is highly innovative, offering significant improvements in brightness, photostability, and size tunability over current taggants. The multi-scale design of the PUFs, incorporating nanoscale, microscale, and macroscale features, enhances their complexity and security. The experimental results are robust, showing high resilience to UV radiation and ease of detection. The integration of deep-learning models to enhance performance and accessibility is particularly impressive, making complex concepts accessible to a broad audience. The potential applications of this technology in next-generation security devices are vast. The authors effectively highlight the practical implications of their work for general end users. The strategy presented for developing PUFs with multiscale features could set a new standard in the field, paving the way for further research and development.

I have a few comments that should be addressed before publication:

For optical PUFs, the challenge-response-pairs (CRPs) do not appear to be very high. It would be beneficial to show more detailed distinctions based on incident angle or wavelength. Additionally, providing a table comparing these CRPs with those reported in recent papers would significantly highlight the manuscript. It appears that some of the latest papers are missing from the citations.

We are sorry for the confusion here. The number we offered in the manuscript is the number of bits. To make it as the number of CRPs, there will be a base of 2 under the bit number for binary codes. This will end up as a significantly large number.

We agree a comparison table will be helpful. Depending on how these values are reported in the literature, they may give theoretical or practical CRPs. With this in mind, we calculated both CRPs based on an area with a size of 1 mm x 1 mm for comparison. For theoretical CRPs of binary codes:

$$\text{The number of CRPs} = 2^{N_{Px}}$$

Where N_{Px} is the number of pixels in the considered area. The pixel size is $0.167 \mu\text{m} \times 0.167 \mu\text{m}$, therefore, for a 1 x 1 mm image of our three-channel PUF:

$$\text{The number of CRPs} = 2^{3 \cdot (1000 \div 0.167)^2} \approx 10^{32381584}$$

The theoretical encoding capacity is known to be readily reduced by small variations in the readout. To minimize the influence from these variations in practical applications, we found that packing multiple pixels as a unit for digitalization instead of a single pixel was beneficial. Specifically, a unit for green and red channels is 10×10 pixels, while a unit for blue channel is 50×50 pixels. This parameter has been proven to show high reproducibility (0.94432 ± 0.0181) when 100 PUFs were tested. In this case,

$$\text{The number of CRPs} = 2^{(1000 \div 0.167 \div 10)^2} \times 2^{(1000 \div 0.167 \div 10)^2} \times 2^{(1000 \div 0.167 \div 50)^2} \approx 10^{220322}$$

The table below shows that our PUF devices have high CRPs when comparing with recent literature:

Table. R2-1 Comparison between the proposed PUF device and recently reported PUFs from the literature.

Type of device	Materials	Bits	Number of CRPs	Size	Literature
Electrical PUF	Metal oxide thin film	28-bit	Not mentioned	20 × 20 mm ²	Nat Commun 11 , 5543 (2020)
Concealable memristor PUF	Hafnium oxide	128-bit	Not mentioned	427 x 352 μm ²	Sci. Adv. , 8 , eabn7753(2022)
Electromagnetic PUF	CPAL devices	256-bit	2 ²⁸⁹	Not mentioned	Sci. Adv. 9 , eadg7481(2023)
Optical PUF	Quantum dots	Not mentioned	4.7 × 10 ²⁰²	Around 80 x 80 μm ²	Nat Commun. 10 , 2409 (2019)
Optical PUF	Carbon dots	Not mentioned	10 ^{63,593}	700 x 700 μm ²	Nat Nanotechnol. 18 , 1027–1035 (2023)
Optical PUF	Gold	Not mentioned	10 ³⁴⁸	100 x 100 μm ²	Nat Commun. 14 , 2185 (2023)
Optical PUF	Silk fibers	Not mentioned	2 ⁷⁶⁸	1 x 1 cm ²	Nat Commun. 13 , 247 (2022)
Optical PUF	Planar spin glass	Not mentioned	10 ⁵⁵	600 x 600 μm ²	Adv. Mater. 35 , 2303077 (2023)
Optical PUF	Organic Crystal	Not mentioned	10 ⁷⁰⁰⁰	1 x 1 mm ²	Adv. Mater. 33 , 2102542 (2021)
Optical PUF	Lanthanide luminescence	Not mentioned	6 × 10 ¹⁰⁴	30 x 30 μm ²	Sci. Adv. 4 , e1701384 (2018)
Optical PUF	Semiconducting polymer nanoparticles	731471	2⁷³¹⁴⁷¹ ≈ 10²²⁰³²²	1 x 1 mm²	This work

We have updated the description in the manuscript as following:

Even with a small area (1 mm²), we found that an immense encoding capacity of *ca.* 10²²⁰³²² can be achieved from the fluorescence channels alone (detailed calculation and a comparison to the literature can be found in Supplementary Note 1 (supplementary information)).

We added a ‘supplementary note 1’ to the SI, which includes the equations above and the comparison table.

To demonstrate practical applicability, it is necessary to present reliability tests, including temperature, humidity, and aging tests. Observing how key characteristics like the Bit Error Rate (BER) change is crucial.

Thank you for the suggestion. We have performed additional reliability tests with our PUFs stored for one week under elevated temperature (40 °C), low temperature (-20 °C) and high humidity (85% humidity). We recorded the patterns of 100 PUFs on day 0 as reference responses. The BER was calculated by the following equation:

$$BER = \frac{1}{k} \sum_{i=1}^k \frac{1}{T} \sum_{l=0}^T \frac{HD(R_i^{Ref}(n), R_i^l(n))}{n}$$

where $R_i^{Ref}(n)$ is the n -bit reference response at Day 0 from the i^{th} PUF, k is the total number of PUF patterns and T is the number of trials.

As shown in Figure R2-1, most of the PUFs show low bit errors after they were treated under different temperature or humidity for one week. All PUFs show reliabilities above 90%, suggesting the excellent performance of the devices under prolonged exposure to these conditions.

Figure R2-1. Bit errors and reliabilities of PUF samples under various conditions.

We have added this data to the end of Figure 5, please the response to the next question for the full figure 5.

We have also added the details of the calculations to a new ‘Supplementary Note 2’:

Supplementary Note 2

The bit-error-rate (BER) was calculated by the following equation:

$$BER = \frac{1}{k} \sum_{i=1}^k \frac{1}{T} \sum_{l=0}^T \frac{HD(R_i^{Ref}(n), R_i^l(n))}{n}$$

where $R_i^{Ref}(n)$ is the n -bit reference response at Day 0 from the i^{th} PUF, k is the total number of PUF patterns and T is the number of trials.

We have also added an additional paragraph in the main text:

Aging tests were then conducted on the multichannel devices. SPNs embedded in photoresists were sustained at an elevated temperature (40 °C), low temperature (-20 °C) and high humidity (85%) for one week. Images were then compared to those taken initially to measure the bit error rate. Fig. 5e shows the bit error rate calculation and values obtained for 100 PUFs in each device. In all cases the PUFs show reliabilities above 90% (Fig. 5f) suggesting the excellent performance of the devices under prolonged exposure to these conditions. Details of calculations can be found in Supplementary Note 2, Supplementary Information.

The machine learning results seem to disrupt the overall narrative of the manuscript. It would be better to move these results to the supplementary information. The figure set is also somewhat difficult to understand and could be improved for clarity.

We appreciate this comment and we think it is a good suggestion. Therefore, we have changed the main text to address this comment by moving the original figure 5c (feature matching by machine learning) to a new 'Figure S16'. In this new SI figure we show the image before and after the match weighting process is applied as an example (see below). Meanwhile, we added a new scheme to the first panel of Figure 5 to show how the PUF labels work. This should help improve the clarity of the whole process. The results from temperature and humidity tests will also be included in the updated Figure 5 following the UV test. In this case, a more general stability analysis is discussed to demonstrate practical applicability.

Please see below for a revised Figure 5, including an edited caption:

Figure 5. Authentication process, practical application and stability test. a) Workflow for the generation of challenge and response pairs, and machine learning assisted validation. b) The optical PUF devices can be attached to different products including circuit boards, medicine packaging, and paper certificates for practical applications. c) Photostability test of the PUF devices. UV wavelength: 365 nm. UV intensity: 12 W/cm². Green channel: E_x 405 nm, E_m 470-550 nm, 20% laser power, 100% gain. d) Match ratios between UV irradiated PUF and the original PUF, based on the feature detecting

results from the LoFTR algorithm. **e)** Bit error rate for each PUF under the storage conditions; **f)** reliability for each PUF in each storage condition. Values compared to images taken at day 0 and after one week. 100 PUFs were recorded per storage condition.

See below new Figure S16:

Supplementary Figure 16. An example of feature matching showing the images before and after applying the LoFTR algorithm.

We also added an addition sentence to the main text to address this change:

Figure S16 shows the feature matching process using the model using one PUF pair as an example, the remaining pairs can be found in Fig S17-19.

While demonstrating scalability might be challenging for an initial study on PUFs, it would be beneficial to discuss the potential for scalability to some extent.

We agree that scalability is important for future applications. We chose spin-coating to produce thin-films, which is unfortunately not-scalable due to its batch-to-batch nature. However, it is well documented that high-quality thin films can be produced with continuous production via dip-coating, roll-to-roll and slot-die, to give a few examples (<https://www.ossila.com/pages/solution-processing-techniques-comparison>). Photolithography is a technique that has been shown to be scalable with the mass production of semiconducting microchips.

We have changed the text to include this brief discussion:

Through exploration of surfactants and anti-solvents, we discovered that small SPN aggregates could be formed and embedded in a photoresist thin-films. Spin-coating was used to produce thin-films here but in future we envisage that more scalable techniques such as roll-to-roll, slot-die or dip-coating could be readily employed.

The process of measuring and obtaining the PUF key is somewhat omitted. Providing more detailed information would be helpful for the readers.

We thank the reviewer for this feedback. We have edited the text to expand on the way in which the PUF key is measured and obtained in the methods section and altered the main text to accommodate for this. Please see our edits below:

“To ensure high reproducibility, we use 10 x 10 pixels as a unit for binarization instead of just a single pixel (the size of a single pixel is 0.167 μm x 0.167 μm). In general, we found PFO SPNs to be smaller than both F8BT and F8BT-red NPs (Fig. S6). As such, we therefore united a larger area (50 x 50 pixels) to enhance the reliability. We then combined three challenge-response pairs together with the final digitized key resulted in 216 bits per microspot (Fig. 3e).”

“Generation of the PUF keys

The raw fluorescent images were normalized and converted to a binary bitmap28 (Fig. 3d). We then use a binning process to remove 50 pixels from the edges of microspots. To ensure high reproducibility, we use 10 x 10 pixels as a unit for binarization instead of just a single pixel (for F8BT and F8BT-Red) and 50 x 50 pixels for PFO SPNs. The intensity of each unit was calculated and upon exceeding the median, the unit was assigned to 1-bit. We then combined three challenge-response pairs together with the final digitized key resulted in 216 bits per microspot, with 100 bits from the red and green channels respectively and 16 bits from the blue channel.”

Reviewer 3

The authors have fabricated PUFs by using semiconducting polymer nanoparticles (SPNs) as fluorescent taggants. The PUFs consist of designs from nanoscale to macroscale. With the assistance of a deep-learning model the authors show that the PUFs have both near-ideal performance and accessibility for general end users, offering a strategy for next-generation security devices. This work may help advance the development of optical PUFs, which is of great current interest. I would like to ask the authors to consider the following points.

We appreciate the reviewers praise of our work. We have addressed each comment in full below.

1. The authors begin with the screening of fluorescent nanoparticles. This is in fact quite challenging given the large quantity of fluorescent nanoparticles. Only 4 types of fluorescent nanoparticles are compared in the current work. Other typical fluorescent nanoparticles such as recently reported silicon nanoparticles (Nature Communications 15, 3203 (2024). DOI10.1038/s41467-024-47479-y) are missing. Please consider a better strategy for the comparison (e. g., combining own experiments with published results in literature).

We agree that it is challenging to compare the vast array of fluorescent nanoparticles reported in the literature.

A barrier we faced was making an objective comparison between literature reported photostability and particle brightness. This is because the optical equipment, sample preparation and experimental set-up between research groups will vary quite considerably. Unfortunately, important parameters such as the illumination power (per unit area) are also often not included in such studies, which are vital for such comparisons. As a result, we opted to purchase a selection of popular commercially available fluorescent nanoparticles and compare them under identical experimental conditions. We would not feel confident or comfortable making any claims to how our materials compare to those reported in the literature.

We have edited the text to reflect on this, and we have also added the newer citation – which was published after the experimental work was completed.

“Transition metal quantum dots (Qdots), perovskite nanocrystals, organic dyes, and carbon dots are the fluorescent taggants most commonly used in the development of optical PUF devices. More recently, silicon Qdots have also been shown as an alternative promising optical PUF taggant.¹⁵”

“We opted to compare the brightness and photostability of materials through an experimental approach under identical conditions, as accurately comparing literature reported values with confidence is challenging. Therefore we initially screened carbon dots, SPNs (poly(9,9-dioctylfluorene-alt-benzothiadiazole, (F8BT)), Qdots (CdSe:ZnS) and perovskite nanocrystals (CsPbBr₃), which have gained the most attention in the field as fluorescent taggants.”

2. Could the authors more clearly justify the choice of F8BT NPs among all SPNs?

F8BT is a popular green/yellow emissive polymer that has the great quality of being commercially available through major suppliers, excellent photoluminescence quantum yield reported and is very soluble in THF. It was also chosen as the red fluorescent polymer, 'F8BT-red' (poly[(9,9-dihexylfluorene)-co-2,1,3-benzothiadiazole-co-4,7-di(thiophen-2-yl)-2,1,3-benzothiadiazole]), is structurally very similar to F8BT.

We could have chosen another polymer such as PFO-co-MEH-PPV which has similar properties to F8BT and there is no reason to suggest that this would not work just as well as F8BT.

We have edited the text to add this justification:

“F8BT was the semiconducting polymer of choice due to its prevalence in the literature, high reported brightness, photostability and good solubility in organic solvents.²⁹”

3. The optical signal is collected by using confocal laser scanning microscopy. Please comment on the practical use of this technique in terms of convenience, portability, etc.

Confocal microscopes were used in this work due to the practical availability in the lab. They are unfortunately rather large and expensive which makes them less practical for product authentication. This comment inspired us to explore a cheaper, more portable alternative. Here, we showcase an affordable commercial-grade USB fluorescent microscope (Dino-Lite, ~£500) to image the fluorescent PUF. The photos were taken on a black surface to minimize reflection. We have tested the same process flow using images with different exposure times (Figure a and b below) and different regions (Figure a and c). Analysis of two identical images (same region and same exposure) shows a high number of matching regions (4256 in Figure d). Benchmarking with this result, analysis among two different exposure times shows similar result (Figure e), indicating the robust analysis pipeline to level the discrepancy in brightness. Expectedly, analysis shows only negligible matching regions from images taken in different regions (Figure f). This result verified that we can also use affordable USB fluorescent microscope for counterfeiting with our developed techniques.

We have added this additional supplementary figure to the end of the manuscript:

Supplementary Figure 20. Images recorded by a USB microscope and the following feature matching analysis with LoFTR algorithm. Patterns imaged in a) long exposure time (1s) in area 1, b) short exposure time(1/4s) also in area 1, c) long exposure time (1s) in area 2. (d) Feature matching between the two images obtained from same exposure time and area. e) Feature matching between the two images obtained from same PUF but different exposure times. f) Feature matching between the two images obtained from different PUFs recorded with same exposure time. g) Photograph of USB microscope with a ten pence coin for size comparison.

We have also edited the manuscript text to address this:

“CLSM was employed throughout this work to illustrate the use of SPN taggants. We wanted to investigate the use of a smaller, portable and lower cost fluorescent reader which would be more appropriate for real-world applications. Here, we implemented a USB green-fluorescent microscope

(Dino-Lite Digital Microscope, ca. £500) to image and authenticate the PUFs. Figure S20a-c shows fluorescent images taken with the portable microscope, in which SPN aggregates can clearly be seen. Employing the same LoFTR algorithm as described above showed excellent matching scores between identical images (Figure S20d) and images with varied exposure time (Figure S20e). Furthermore, low matching scores were found for images in two different PUF regions (Figure S20f). An image of the microscope can also be found in Figure S20g. This illustrates that the authentication of the fluorescent PUFs could be achieved with a portable microscope, making these PUFs practical for real-world product authentication.”

4. Please tell the exact size of a pixel in the manuscript.

The pixel size is 0.167 μm x 0.167 μm .

We have edited the text:

To ensure high reproducibility, we use 10 x 10 pixels as a unit for binarization instead of just a single pixel (the size of a single pixel is 0.167 μm x 0.167 μm).

REVIEWERS' COMMENTS

Reviewer #1 (Remarks to the Author):

I am totally satisfied with the revision, and please publish as it is.

Reviewer #2 (Remarks to the Author):

The revised version is well-organized and ready for publication. One suggestion would be to carefully review the challenge-response pair (CRP) parameter once more, given its significance. It may be better to move the table from the supplementary information (SI) to the main table.

Reviewer #3 (Remarks to the Author):

The authors have clearly improved the manuscript by addressing the reviewers' comments. I would like to recommend the publication of this revised manuscript.

Point-by-point response to the reviewers' comments

We have made final adjustments to the manuscript to address the final reviewer comments. Reviewer #1 and #3 had no further suggested edits. Reviewer #2 made the following comment:

“The revised version is well-organized and ready for publication. One suggestion would be to carefully review the challenge-response pair (CRP) parameter once more, given its significance. It may be better to move the table from the supplementary information (SI) to the main table.”

To address this, we have moved the table from ‘Supporting Note 1’ and moved it to the main text, and made small alterations to the text to address this. The reference list was updated accordingly.

We attach the updated manuscript for publication.